# Discovery and Comprehensive Characterization of Novel Circular RNAs of the Apoptosis-Related *BOK* Gene in Human Ovarian and Prostate Cancer Cells, Using Nanopore Sequencing

**DOI:** 10.3390/ncrna9050057

**Published:** 2023-09-24

**Authors:** Christos K. Kontos, Despina Hadjichambi, Maria Papatsirou, Paraskevi Karousi, Spyridon Christodoulou, Diamantis C. Sideris, Andreas Scorilas

**Affiliations:** 1Department of Biochemistry and Molecular Biology, Faculty of Biology, National and Kapodistrian University of Athens, 15701 Athens, Greece; depihad@biol.uoa.gr (D.H.); papatsir@biol.uoa.gr (M.P.); pkarousi@biol.uoa.gr (P.K.); dsideris@biol.uoa.gr (D.C.S.); ascorilas@biol.uoa.gr (A.S.); 2Fourth Department of Surgery, University General Hospital “Attikon”, National and Kapodistrian University of Athens, 12462 Athens, Greece; spyridon.christodoulou@yahoo.gr

**Keywords:** circRNAs, alternative splicing, transcriptomics, third-generation sequencing, miRNAs, reproductive cancers

## Abstract

CircRNAs have become a novel scientific research hotspot, and an increasing number of studies have shed light on their involvement in malignant progression. Prompted by the apparent scientific gap in circRNAs from apoptosis-related genes, such as *BOK*, we focused on the identification of novel *BOK* circRNAs in human ovarian and prostate cancer cells. Total RNA was extracted from ovarian and prostate cancer cell lines and reversely transcribed using random hexamer primers. A series of PCR assays utilizing gene-specific divergent primers were carried out. Next, third-generation sequencing based on nanopore technology followed by extensive bioinformatics analysis led to the discovery of 23 novel circRNAs. These novel circRNAs consist of both exonic and intronic regions of the *BOK* gene. Interestingly, the exons that form the back-splice junction were truncated in most circRNAs, and multiple back-splice sites were found for each *BOK* exon. Moreover, several *BOK* circRNAs are predicted to sponge microRNAs with a key role in reproductive cancers, while the presence of putative open reading frames indicates their translational potential. Overall, this study suggests that distinct alternative splicing events lead to the production of novel *BOK* circRNAs, which could come into play in the molecular landscape and clinical investigation of ovarian and prostate cancer.

## 1. Introduction

Cancers of the reproductive organs, including human ovarian and prostate cancer, account for a big percentage of cancer morbidity and mortality worldwide, while their incidences have gradually increased in many countries [1]. According to the American Cancer Society, an estimated 268,490 men and 19,880 women were diagnosed with prostate and ovarian cancer, respectively, in 2022, with estimated mortality percentages around 12% and 64%, respectively. Several attempts have been made towards the elucidation of the molecular basis of these malignancies, in order to gain insights into the events that lead to carcinogenesis and suggest successful therapeutic regimens.

Apoptosis is a significant component of numerous physiological processes, including normal cell turnover, proper tissue development, and maintenance of organismal homeostasis [2]. Impairment of apoptosis followed by increased cell survival results in uncontrolled metabolism and tumor formation [3,4]. The BCL2 protein family, which consists of both pro- and anti-apoptotic members, is a critical player in the regulation and execution of the intrinsic apoptotic pathway that dictates cell fate by balancing the decision between cell death and survival [5]. Particularly, in prostate cancer, the molecular alterations in most cancerous cells lead to numerous apoptosis-blocking strategies during the stages of progression from normal epithelial cells to androgen-dependent tumor cells, and further onto malignant androgen-independent tumor cells [6]. In ovarian cancer, malignant cells gain a survival advantage that frequently equates to a more resistant phenotype, resulting from direct alterations of the levels of pro-apoptotic proteins. These apoptotic-resistant ovarian cancer cells are significantly influenced by BOK, which probably contributes to the high recurrence rates of chemo-resistant disease [7].

*BOK*, also known as BCL-2-related ovarian killer, is an insufficiently understood member of the BCL2 family [8]. The term “ovarian killer” was coined since *BOK* was first found in ovarian tissue. Further investigation has revealed that it is also present in other tissue types, but that reproductive tissues are where it is prominently expressed [9,10]. Although BOK has been reported to typically exert a pro-apoptotic function, the presence of a BCL2-homology 4 (BH4) domain in its structure (a typical characteristic of pro-survival proteins), creates an ambiguity regarding the role of this molecule [11]. Current knowledge of ovarian cancer provides strong indications that not only establish BOK as an inducer of apoptosis in a BAX- and BAK1-independent manner, but also suggest that BOK can significantly influence the apoptotic response to chemotherapeutic drugs in ovarian carcinoma cells [12]. Alternative splicing as a mechanism can produce *BOK* transcript variants that might contribute to its diverse functions in cancer. However, there have been no studies so far regarding the potential of *BOK* to produce circular transcripts, either in normal or pathological conditions.

CircRNAs constitute a subpopulation of non-coding RNAs, characterized by a covalent loop structure, which are produced by back-splicing [13]. Driven by the progress in high-throughput sequencing and emerging bioinformatics approaches, it has recently been confirmed that circRNAs are widely expressed in mammalian transcriptomes and intricately regulate gene expression and miRNA function [14,15]. Despite their multifaceted roles in cellular physiology, circRNAs have been linked to the pathogenesis and progression of several human diseases [15,16], including reproductive malignancies, as exemplified by circ_0085494 and hsa_circ_0063329 in prostate cancer cells [17,18,19], and circ-MUC16, circRNA-MYLK, and circRNA-UBAP2 in ovarian cancer [20,21,22]. However, it is still unclear how circRNAs regulate ovarian and prostate cancer onset and progression; investigating circRNAs derived from an apoptosis-related genes like *BOK* in prostate and ovarian cancer cells allows us to explore potential connections between circRNAs and apoptosis in these cancer types.

For the elucidation of circRNA structure and alternative splicing events, previous studies have presented a workflow using algorithms based on short sequencing reads [23]. However, those approaches possess several limitations, including low detection sensitivity as a result of errors in the assembly of short-read sequencing reads [24,25]. Third-generation sequencing with nanopore technology overcomes most of these limitations by enabling the production of long sequencing reads that provide full-length circRNA sequence reconstruction and detailed characterization of circRNAs [26,27,28,29].

Based on the above, the primary focus of this research study was the identification of novel *BOK* circRNAs deriving from alternative splicing events, in ovarian and prostate cancer cells, a rather unexplored area of scientific inquiry. In addition, this is the first study that utilized a targeted approach for the *BOK* gene, significantly increasing the sensitivity for circRNA identification. For this purpose, targeted third-generation sequencing with nanopore technology was implemented. Overall, this approach benefited from the advent of high-throughput sequencing and an extensive data-processing bioinformatics pipeline and led to the discovery of 23 novel *BOK* circRNAs in ovarian and prostate cancer cells. Besides the revelation of the full-length sequences of *BOK* circRNAs, their putative interactions with miRNAs and their coding potential were also examined.

## 2. Results

### 2.1. Identification and Characterization of Novel BOK circRNAs in Ovarian and Prostate Cancer Cell Lines

Our experimental approach led to the discovery of 23 novel *BOK* circRNAs, which were submitted to GenBank of NCBI (Table 1). Circ-BOK-10 and circ-BOK-19 were previously deposited into the circAtlas 3.0 database as well [30], with circAtlas IDs hsa-BOK_0011 and hsa-BOK_0016, respectively. All novel circRNAs consist of one to four already annotated *BOK* exons, generating both single-exon and multi-exon circular transcripts. The circRNA length varies between 189 and 904 nt. The annotated reference sequences of the novel circRNAs are shown in the Appendix A.

The majority of the detected circRNAs were multi-exonic, comprising either the annotated *BOK* exons or shorter versions of them. Moreover, extended *BOK* exons were found in single-exon circRNAs (circ-BOK-9, circ-BOK-16, circ-BOK-12, and circ-BOK-15), and this way intronic regions of the gene were incorporated into its circular transcripts. Both 5′ or 3′ ends of *BOK* exons could be extended or truncated. We also observed that exon 2 was rarely represented in the novel circRNA sequences. In addition, the splicing of the non-coding exon between the 1st and 2nd coding exons—a rather uncommon exon that was previously found mainly in expressed sequence tags (ESTs)—was detected in circular *BOK* transcripts expressed only in DU 145 and MDAH-2774 cells (Figure 1).

Furthermore, interesting facts about the back-splice junction emerged. Most back-splice sites were non-canonical, meaning that there was neither GT at the donor site nor AG at the acceptor site. In fact, the generation of multiple distinct *BOK* circRNAs is mainly the result of various back-splice sites and not of major differences in the exonic structure or the forward-splicing events. Moreover, back-splicing may occur between exonic and/or intronic regions that share a short sequence identity. It cannot be deduced whether this identical sequence belongs to the end of the downstream exon (donor) or the start of the upstream exon (acceptor) of the back-splice junction.

### 2.2. Expression Analysis of the Novel BOK circRNAs

The novel *BOK* circRNAs are characterized by distinct expression patterns among the seven cancer cell lines (Table 1). In particular, 18 *BOK* circRNAs were detected in the ovarian cancer cell lines and 9 *BOK* circRNAs in prostate cancer cell lines. A representative read proving the expression of each circRNA in the respective cell line(s) is provided in the Appendix A, while the keywords used for the computational analysis of our nanopore sequencing data and the number of representative reads thus emerging in each cell line are shown in Appendix A.

Although we did not detect an obvious cancer-specific circRNA expression pattern, we noticed that the features of each cancer cell line may influence the expression of circRNAs, to some extent. For instance, circ-BOK-10 was exclusively expressed in LNCaP and OVCAR-3, an androgen- and an estrogen-dependent cancer cell line, respectively. Moreover, circ-BOK-20 was detected in two out of three estrogen-dependent cell lines (Ishikawa and OVCAR-3), while it was not detected in the estrogen-independent MDAH-2774 cell line. Additionally, although most *BOK* circRNAs are expressed in only one cell line, their expression pattern does not appear to correlate with the epithelial subtypes of ovarian cancer. For instance, circ-BOK-20 is expressed in cells from both endometrioid adenocarcinoma and high-grade ovarian serous adenocarcinoma. Interestingly, out of the full catalog of the identified circRNAs, only circ-BOK-17 was detected in the DU 145 prostate cancer cell line.

Additionally, we noticed that specific genomic regions of *BOK* are differentially represented among the cell lines. Specifically, extensions of the known *BOK* exons were mainly present in ovarian cancer cell lines. Regarding the non-coding exon between the first and second coding exons, it is highly represented in the DU 145 prostate cancer cell line, while shorter versions of it are also represented in the MDAH-2774 ovarian cancer cell line (Figure 2).

### 2.3. Putative Interactions of BOK circRNAs with miRNAs

Putative interactions between the novel *BOK* circRNAs and miRNAs were evaluated *in silico* using the custom prediction tool of miRDB. By employing this publicly available tool, potential miRNA target sites were reported in all identified *BOK* circRNAs (Appendix A). Out of these miRNAs, we selected those with high prediction target scores (>80). Through this selection, 12 miRNAs were predicted to bind to novel *BOK* circRNAs and are listed along with their prediction score and their binding motifs in Table 2. Interestingly, miR-3158-5p, miR-466, and miR-214-5p are involved in reproductive cancers onset and progression [31,32,33]. In addition, all five miRNAs that were predicted to interact with circ-BOK-15 were observed to bind to the intronic region present in this circRNA.

### 2.4. Prediction of the Translational Potential of BOK circRNAs

We further investigated the presence of open reading frames (ORFs) in our circRNA dataset. Indeed, we found 16 *BOK* circRNAs with putative ORFs using the web tool ORF finder of the NCBI (Table 3). Although there were no internal ribosome entry sites (IRES) in the sequences of these circRNAs, the existence of putative m^6^A modifications in the sequences of these circRNAs was confirmed. In particular, 15 circRNAs that possess a putative ORF were also predicted to possess m^6^A sites as well (Table 3); however, no m^6^A sites were predicted in the circ-BOK-9 sequence. Interestingly, six of them were predicted to possess m^6^A modification sites with very high confidence.

## 3. Discussion

The precise and accurate identification of circRNA sequences is vital for a comprehensive understanding of circRNA functions and mechanisms. These molecules have rightfully been in the spotlight in recent years, and the ongoing elucidation of their role in ovarian and prostate cancer onset and progression has established circRNAs as potent disease regulators and biomarkers [34]. This can be especially informative for genes with a proven role in reproductive malignancies, such as *BOK*. Recent studies point towards a wide complexity of *BOK* in cancer, which can be attributed to the variety of alternative transcripts of this gene. Although most of the studies so far paid more attention to alternative splicing as a key modulator of mRNA production, increasing evidence has shown that alternative splicing of ncRNAs is even more common [35,36,37]. Based on these, we aimed to explore the alternative circular transcripts of *BOK* in ovarian and prostate cancer in such depth that could be elucidated only by a high-throughput sequencing approach [38].

The circular transcriptome assembly during the RNA sequencing data analysis is complex and challenging, mainly due to the high sequence similarity with linear cognates, especially with short-read approaches [39]. In this study, we chose to implement a targeted long-read sequencing pipeline to identify novel circRNAs of the *BOK* gene and decipher its transcriptional landscape in reproductive cancers. This methodology circumvents the length restriction of 200 nt near the back-splice junction of second-generation paired-end sequencing approaches, which is insufficient to read the majority of circRNAs. In fact, many novel *BOK* circRNAs had a length of more than 400 nt, and we would not have been able to identify these sequences through different approaches.

During the annotation of the novel *BOK* circRNAs, intriguing features of the back-splice junction also emerged. The exons that participate in the back-splice junction were in most cases truncated forms of known *BOK* exons. Moreover, most back-splice sites were non-canonical; these non-canonical splicing events, particularly when they involve transposable element-derived sequences, are a significant source of newly emerging transcripts during evolution [40]. Non-canonical splicing events are frequent in cancer because they sidestep cellular quality control systems and compromise the integrity of the transcriptome [40,41]. Thus, it might be likely that specific novel *BOK* circRNAs with non-canonical splicing sites disrupt normal gene expression and alter the ratio of pathological to normal transcript variants in malignant conditions. Another interesting aspect of the back-splice junction of *BOK* circRNAs is the sequence identity that was usually observed in exonic and/or intronic regions forming the back-splice junction. This finding seems to be characteristic of circRNAs deriving from other genes as well. For example, novel *PRMT1* circRNAs in breast cancer and *BCL2L12* circRNAs in colorectal cancer share this feature, where several nucleotides at the exons forming the back-splice junction are identical [38,39,42].

Notably, our research reveals that these novel *BOK* circRNAs exhibit cell line-specific expression profiles, with some circRNAs being unique to particular cancer cell lines. For instance, circ-BOK-10 demonstrates exclusive expression in LNCaP and OVCAR-3, cell lines that are androgen- and estrogen-dependent, respectively, suggesting a potential link between hormone dependency and circRNA expression. Furthermore, the detection of circ-BOK-20 in estrogen-dependent cell lines but its absence in an estrogen-independent cell line underscores the potential involvement of hormonal regulation in circRNA expression. Moreover, the observation of specific genomic regions of *BOK* being differentially represented in various cell lines underscores the dynamic nature of *BOK* gene expression and warrants further investigation to elucidate the functional significance of these variations in different cancer contexts.

Additionally, the use of exon skipping and extended *BOK* exons in the novel circRNA structure was also prevalent in our dataset. These events are in agreement with similar studies that report alternative splicing events in circRNAs [43,44]. Interestingly, the validation of the existence of the non-coding *BOK* exon between the first and second coding exons—which so far has been an outcome of computational analysis of expressed sequence tags (ESTs)—is evidenced by our results, considering that it participates in the structure of three novel circRNAs either as a full-length exon or as a 5′-truncated one. This finding increases the scientific questions around the complexity and purpose of alternative splicing events of *BOK*. Moreover, four single-exon *BOK* circRNAs were identified in our dataset. Interestingly, these circRNAs comprise extended *BOK* exons, which were not found either in multi-exonic circRNAs, or linear transcripts of the gene. According to the relevant literature, single-exon circRNAs derive from unusually long exons while multi-exon circRNAs are mostly generated from exons of regular length [45], which was prevalent in our dataset as well. These single-exon circRNAs underscore the diversity in the structural composition of circRNAs and suggest that circRNA biogenesis is not limited to complex splicing events involving multiple exons but can also occur from single exons. However, while circRNAs are well-known for miRNA sponging and protein binding, the functions of single-exon circRNAs may differ from those of multi-exon circRNAs and remain to be elucidated. Overall, the alternative splicing of *BOK* pre-mRNA produces a variety of transcripts and is expected to result in protein isoforms with a different pattern of BCL2-homology domains or in transcripts with distinct regulatory functions. The same could apply to the intense alternative splicing events found in circular *BOK* transcripts, implying a potentially different role of the detected circRNAs in the cell. This fact would be interesting to examine in parallel with normal ovarian and prostate cells.

Our data enabled a full-length sequence analysis of circRNAs containing previously intronic regions of *BOK*. This finding adds to the evidence that intronic regions within this gene are functionally active and present in circular transcripts. This observation suggests that competitive splicing between circular and linear *BOK* transcripts occurs [46]. It is worth mentioning that circRNAs with intronic sequences are frequently discovered in the nucleus and, in some situations, lead to higher levels of expression of their host gene. Interestingly, circ-BOK-15 may sequester miR-545-3p and miR-4700-5p through binding sites in the intronic region that is incorporated in this circRNA, meaning that these miRNAs can only interact with this circRNA among linear and other circular transcripts of *BOK*. Furthermore, miR-545-3p is proven to be sponged by other circRNAs and affect cancer progression through signaling regulation, as well as proliferation, migration, and invasion of cancer cells [47,48,49]. These findings render circ-BOK-15 a promising candidate for further functional experiments, especially the circ-BOK-15/miR-545-3p regulatory axis.

Moreover, the functional in silico analysis of this study sheds light on the potential interactions of the novel *BOK* circRNAs in ovarian and prostate cancer, particularly focusing on their miRNA sponging potential. Potential miRNA target sites were predicted in all *BOK* circRNAs. Interestingly, some of these miRNAs with high prediction target scores, including miR-3158-5p, miR-466, and miR-214-5p, have been previously implicated in the onset and progression of reproductive cancers [31,33,50]. By sequestering miRNAs involved in reproductive cancers, *BOK* circRNAs might play critical roles in altering the expression levels of tumor-promoting and tumor-suppressing genes that are relevant to these malignancies. 

In addition, 15 out of the 16 *BOK* circRNAs possessing putative ORFs were predicted to harbor m^6^A sites as well, strengthening the possibility that these transcripts might be translationally active. This modification is associated with the initiation of circRNA translation in a cap-independent manner, and a single m^6^A site is sufficient to drive translation initiation [51]. Moreover, alternative translation initiation sites reside in several *BOK* exons. The presence of putative ORFs and m^6^A sites in *BOK* circRNAs suggests that they could produce functional peptides or proteins. This contributes to a better understanding of the functional diversity of circRNAs, which were previously assumed to be non-coding RNA molecules; however, these findings are so far only based on computational predictions and require further experimental validation.

Although the experimental pipeline that was followed in this study allowed several unknown circRNA characteristics to come to the surface, there are a few limitations as well. Firstly, the relatively high error rate of nanopore sequencing reads influences the accuracy of circRNA detection, and some *BOK* circRNAs could not be annotated due to mismatches at the back-splice junction. Additionally, our data does not distinguish between nuclear and cytoplasmic circRNAs, and it is yet unknown how much intron retention can influence the compartmentalization of circRNAs within cells or whether it serves any regulatory purposes. Moreover, while our study provides valuable insights into the *BOK* circRNAs in human ovarian and prostate cancer cells, we acknowledge the limitation of not having access to patient samples. 

While our study has unveiled a comprehensive profile of novel circRNAs associated with the apoptosis-related *BOK* gene in human ovarian and prostate cancer cells, it is imperative to highlight potential avenues for future research in this field. Primarily, the validation of our findings in patient samples remains a critical endeavor to assess the clinical relevance of these circular RNAs as diagnostic or therapeutic targets. Obtaining patient samples and conducting further investigations in this context would be essential to bridge the gap between our cell line-based study and clinical applications. Furthermore, functional studies, including siRNA knockdown experiments and mechanistic investigations, are essential to elucidate the precise roles of these circular RNAs in tumorigenesis and apoptosis regulation. Additionally, the exploration of potential interactions between these circular RNAs and other biomolecules, such as proteins or miRNAs, could provide a more comprehensive understanding of their regulatory networks. 

## 4. Materials and Methods

### 4.1. Cell Culture

For the implementation of the current workflow, 4 human ovarian and 3 human prostate cancer cell lines were used. In particular, the MDAH-2774 (ATCC CRL-10303), ES-2 (ATCC CRL-1978), OVCAR-3 (ATCC HTB-161), and Ishikawa (ECACC 99040201) ovarian cancer cell lines, and the DU 145 (ATCC HTB-81), PC-3 (ATCC CRL-1435), and LNCaP (ATCC CRL-1740) prostate cancer cell lines were propagated. The ovarian cancer cell lines are of distinct epithelial subtypes: MDAH-2774 and Ishikawa were established from ovarian endometrioid adenocarcinomas, ES-2 from a poorly differentiated ovarian clear cell carcinoma, and OVCAR-3 from high-grade ovarian serous adenocarcinoma. Regarding the prostate cancer cell lines, DU 145 and LNCaP were established from prostate carcinoma, while PC-3 was established from grade IV adenocarcinoma. All cell lines were cultured following each supplier’s guidelines. Cells were incubated at 37 °C, in a humidified atmosphere with CO_2_ concentration adjusted to 5%.

### 4.2. Total RNA Extraction and Reverse Transcription

Total RNA was isolated from each of the 7 cell lines, when cells reached 80% confluency, using the TRItidy G™ Reagent (AppliChem GmbH, Darmstadt, Germany) according to the manufacturer’s protocol. The purity and concentration of the extracted RNA were evaluated spectrophotometrically at 260 and 280 nm. Next, 2 μg of each RNA sample were reversely transcribed using random hexamer primers (New England Biolabs Ltd., Hitchin, UK), with the M-MLV reverse transcriptase (Invitrogen™, Thermo Fisher Scientific Inc., Waltham, MA, USA). The first-strand cDNA synthesis was conducted as previously described, resulting in 20 μL of total cDNA volume [38].

### 4.3. Primer Designing and Amplification of BOK circRNAs

Two pairs of gene-specific divergent primers were designed for the 2nd, 3rd, and 4th coding *BOK* exons (numbering based on the main *BOK* mRNA, RefSeq ID: NM_032515.5): one pair for the first-round PCR and a second, internal pair for a nested PCR assay (Table 4). First-round and nested PCR assays were carried out for each amplicon, using KAPA Taq DNA Polymerase (KAPA Biosystems Inc., Woburn, MA, USA) in a MiniAmp Thermal Cycler (Applied Biosystems™, Thermo Fisher Scientific Inc., Waltham, MA, USA). The reaction mixture in both assays included 19.4 μL nuclease-free H_2_O, 1X KAPA Taq Buffer, 200 μM of each dNTP, 400 nM of each primer, 0.5 U KAPA Taq DNA Polymerase, and 0.5 μL of cDNA (first-round PCR) or diluted PCR product at a ratio of 1:50 in nuclease-free H_2_O (nested PCR), with a final reaction volume of 25 μL. The thermal protocol was conducted according to the manufacturer’s guidelines: a denaturation step at 95 °C for 3 min, followed by 35 cycles of denaturation at 95 °C for 30 s, an annealing step at 61 °C for 1 min, and an extension step at 72 °C for 30 s. The final elongation was carried out at 72 °C for 1 min. Then, agarose gel electrophoresis of the PCR products was performed and followed the purification of the specific amplicons. The PCR products of each cell line were mixed and purified at equal volumes using spin columns (NucleoSpin Gel and PCR Clean-up Column, Macherey-Nagel GmbH & Co. KG, Düren, Germany). The concentration of the purified products was evaluated using a Qubit 2.0 Fluorometer (Invitrogen™, Thermo Fisher Scientific Inc., Waltham, MA, USA).

### 4.4. Library Preparation and Third-Generation Sequencing with Nanopore Technology

A DNA library for barcoding and adapter ligation was prepared using the Native Barcoding Expansion Kit 13–24 and the Ligation Sequencing Kit (Oxford Nanopore Technologies plc., Oxford, UK), following the manufacturer’s protocols. Briefly, the purified, pooled PCR products derived from each cell line were used as input for the construction of a barcoded library. A Qubit 2.0 Fluorometer was used for the quantification of the barcoded libraries. Nanopore sequencing was performed on the MinION Mk1C platform with the Flongle adapter (Oxford Nanopore Technologies plc., Oxford, UK). A FLO-MIN106D flow cell with R9.4.1 chemistry was used, following the manufacturer’s instructions for priming and loading the flow cell.

### 4.5. Third-Generation Sequencing Data Analysis

The passed raw sequencing reads contained in the FASTQ files were mapped against the human reference genome (GRCh38) with the Minimap2 aligner [52]. Alignment with Minimap2 led to the generation of output SAM files containing the successfully mapped sequencing reads for each barcode. Next, the SAM files were sorted, and the sorted SAM files were used as input to generate BAM files using SAMtools. Mapped sequencing reads were visualized with the Integrative Genomics Viewer (IGV) software (version: 2.16.2) [53].

Besides mapping with Minimap2, the detection of back-splicing events in the FASTQ files was conducted using a series of Perl-based algorithms. Firstly, the “ASDT” (Alternative Splicing Detection Tool) algorithm was used, providing a modified GenBank^®^ file for the *BOK* gene as input [54]. Following this, we used the “ASDT remodeler” algorithm (https://github.com/pkarousi/ASDT_remodeler.git accessed on 15 March 2023) to discern the reads representing *BOK* circRNAs. Then, extended manual annotation was conducted for each cancer cell line (Figure 3). Finally, the “Read catcher” algorithm (https://github.com/pkarousi/Read_catcher.git accessed on 10 June 2023) was used to examine the presence of the novel identified *BOK* circRNAs across the 7 cell lines, based on the sequence of all splice junctions, including the back-splice junction itself (Appendix A). The step-by-step data processing pipeline for the identification of novel *BOK* circRNAs is illustrated in Appendix A.

### 4.6. In Silico Analysis for the Putative Functions of the Novel BOK circRNAs

In silico analysis was performed to predict interactions of the novel *BOK* circRNAs. Particularly, all circRNAs were evaluated using the miRDB custom prediction web tool [55], regarding their ability to sponge miRNAs. The circRNA transcript sequences were submitted and a compilation of miRNAs that could potentially be sponged was provided, as well as a probability score.

Then, further investigation of our circRNA dataset was performed regarding their potential to be translated. For this purpose, the ORF Finder bioinformatics tool (https://www.ncbi.nlm.nih.gov/orffinder accessed on 10 September 2023) was employed to search for putative open reading frames (ORFs) within the nucleotide sequences the novel identified circRNAs. Then, ORFs were queried in all reading frames starting with the ATG codon, and only those ORFs producing putative peptides exceeding 30 amino acids were considered. Then, the prediction of the N6-methyladenosine (m^6^A) modification sites on the *BOK* circRNA sequences that possess a putative ORF followed, via the SRAMP prediction server [56]. The presence of such sites is significant for the prediction of *BOK* circRNAs translational potential, as post-transcriptional methylation is largely implicated, in a cap-independent manner, in the regulation of the circRNA translation process. Particularly, the circRNA sequences were submitted in the SRAMP prediction server in FASTA format, and the analysis of RNA secondary structure was also chosen so that the m^6^A sites would be visualized. Finally, the existence of IRES was also predicted, providing a FASTA file with the sequences of the 17 circRNAs with putative ORFs to the IRESpy bioinformatics tool [57].

## 5. Conclusions

In conclusion, it is evident that *BOK* is a gene with abundant transcriptional activity that generates a wide range of circRNAs, which contain numerous complex alternative splicing events. It would be interesting to investigate the alterations of the expression levels of the identified circRNAs, not only in cancerous cells but also in normal ones, under the effect of therapeutic drugs. Moreover, the potential of *BOK* circRNAs as diagnostic and prognostic molecular biomarkers could be elucidated in the future by assessing their expression in ovarian and prostate cancer patients’ tissue samples. The comprehensive understanding of the function of the novel *BOK* circRNAs function will aid in elucidating the possible need for such a wide range of circRNAs and their involvement in ovarian and prostate cancer.

## Figures and Tables

**Figure 1 ncrna-09-00057-f001:**
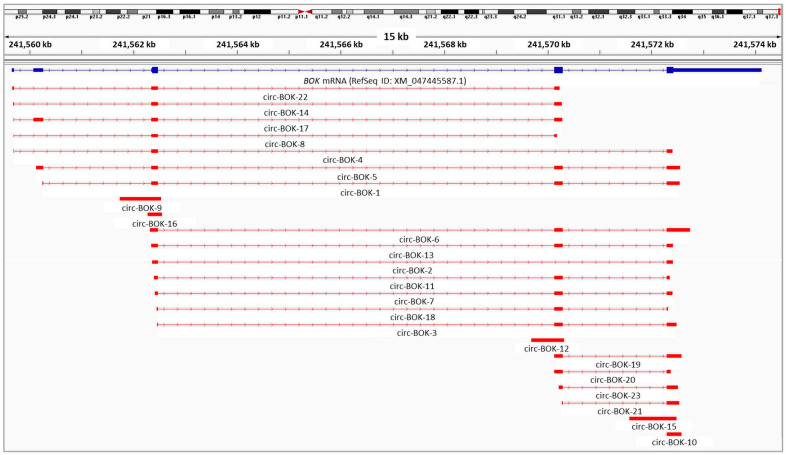
Visualization of the alignment of the novel *BOK* circRNAs identified in the ovarian and prostate cancer cell lines against human chromosome 2, using the Integrative Genomics Viewer (IGV). All circRNAs are depicted starting from the back-splice acceptor site. The visualized BED file is provided in the Appendix A.

**Figure 2 ncrna-09-00057-f002:**
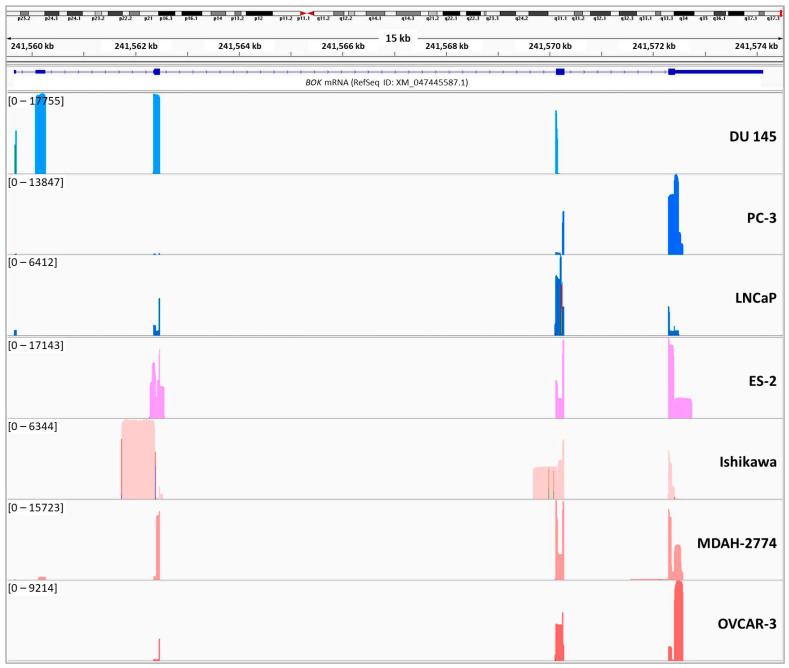
Coverage of genomic sequence of *BOK* in each prostate and ovarian cancer cell line, used in the current study, using the Integrative Genomics Viewer (IGV).

**Figure 3 ncrna-09-00057-f003:**
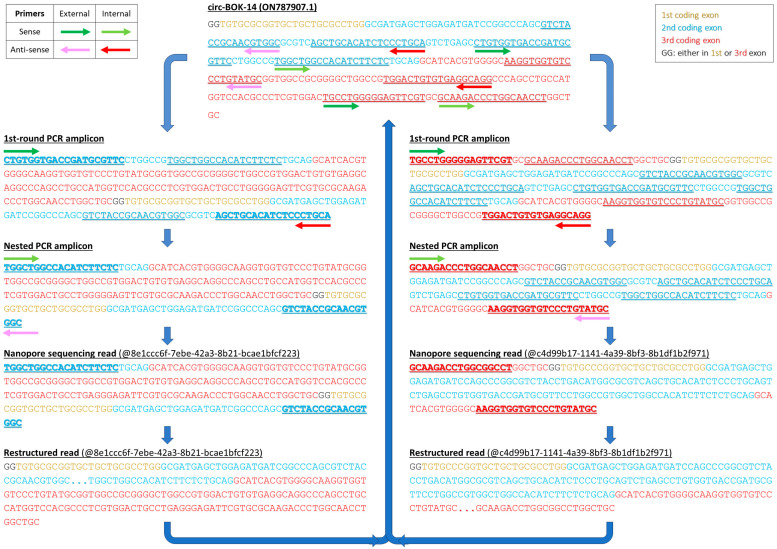
Schematic representation of the methodology followed to deduce the sequence of each circRNA, after targeted nanopore sequencing. The example of circ-BOK-14 is presented; its amplification was achieved using 2 pairs of divergent primers, all of them annealing either in the 2nd or the 3rd coding exon of BOK gene. As a result, circ-BOK-14 is represented by 2 different amplicons. Both ends of these representative reads have been trimmed so as to exclude junk sequence.

**Table 1 ncrna-09-00057-t001:** Expression analysis^1^ of the 23 novel *BOK* circRNAs in prostate and ovarian cancer cell lines, based on our nanopore sequencing data.

*BOK* circRNA	GenBank Accession #	Prostate Cancer Cell Lines			Ovarian Cancer Cell Lines			
		DU 145	LNCaP	PC-3	ES-2	Ishikawa	MDAH-2774	OVCAR-3
circ-BOK-1	ON787895.1	–	–	–	–	–	+	–
circ-BOK-2	ON787896.1	–	–	–	–	+	–	–
circ-BOK-3	ON787897.1	–	+	–	–	–	–	–
circ-BOK-4	ON787898.1	–	–	–	–	+	–	–
circ-BOK-5	ON787899.1	–	–	–	–	–	+	–
circ-BOK-6	ON787900.1	–	–	–	+	–	–	–
circ-BOK-7	ON787901.1	–	–	–	+	–	–	–
circ-BOK-8	ON787902.1	–	+	–	–	–	+	–
circ-BOK-9	ON787903.1	–	–	–	–	+	–	–
circ-BOK-10	ON787904.1	–	+	–	–	–	–	+
circ-BOK-11	ON787905.1	–	–	–	–	–	+	–
circ-BOK-12	ON854799.1	–	–	–	–	+	–	–
circ-BOK-13	ON787906.1	–	–	+	–	–	–	+
circ-BOK-14	ON787907.1	–	–	–	–	–	–	+
circ-BOK-15	ON787908.1	–	–	–	–	–	+	–
circ-BOK-16	ON787909.1	–	–	–	+	–	–	–
circ-BOK-17	ON787910.1	+	–	–	–	–	–	–
circ-BOK-18	ON787911.1	–	–	+	–	–	–	–
circ-BOK-19	OR523823.1	–	–	–	–	–	+	–
circ-BOK-20	ON787912.1	–	–	–	–	+	–	+
circ-BOK-21	OR523824.1	–	+	–	–	–	+	–
circ-BOK-22	ON787913.1	–	–	+	–	–	–	–
circ-BOK-23	OQ291309.1	–	–	+	–	–	–	–

1 “+” denotes presence and “–” denotes absence of the respective *BOK* circRNA.

**Table 2 ncrna-09-00057-t002:** MicroRNAs (miRNAs) that are highly likely to be sponged by *BOK* circRNAs.

*BOK* circRNA	miRNA Binding to circRNA	Prediction Score ^1^	Binding Motif	Seed Location ^2^
circ-BOK-2	miR-3158-5p	81	UCUGCAGA	110
			UCUGCAG	184
circ-BOK-4	miR-3158-5p	81	UCUGCAGA	133
			UCUGCAG	207
circ-BOK-6	miR-4267	81	GAGCUGGA	29
			GAGCUGG	635
circ-BOK-9	miR-6819-5p	87	CACCCCAA	134
			CACCCCA	187
	miR-6737-5p	87	CACCCCAA	134
			CACCCCA	187
	miR-6812-5p	82	CACCCCAA	134, 187
circ-BOK-12	miR-466	90	AUGUGUAA	80
	miR-214-5p	80	ACAGGCA	115
			GACAGGCA	284
circ-BOK-15	miR-545-3p	89	UUUGCUGA	620
	miR-4700-5p	87	UCCCCAGA	191
	miR-3617-3p	83	UGCUGAUA	622
	miR-8089	83	UCCCCAGA	191
	miR-4667-5p	83	UCCCCAGA	191

^1^ The custom prediction tool of the miRDB database was used for the prediction of miRNAs binding to each *BOK* circRNA. The range of prediction score is 50–100; only those with a prediction score ≥80 are shown. ^2^ Each coordinate refers to the respective circRNA sequence.

**Table 3 ncrna-09-00057-t003:** The coding potential of the novel *BOK* circRNAs possessing a putative open reading frame (ORF).

*BOK* circRNA	Putative Coding Region (Start–Stop) (nt)	Length of the Deduced Protein Sequence (aa)	Predicted m^6^A Site Position (nt) ^1^	m^6^A Prediction Confidence Level ^1^
circ-BOK-1	31–435	134	203/248/310	High/High/Very high
	107–211	34		
circ-BOK-2	2–121	39	117	Low
circ-BOK-3	100–306	68	74/119/181	High/High/Very high
circ-BOK-4	25–144	39	140	Low
circ-BOK-5	157–561	134	79/329/374/436	Low/High/High/Very high
	233–337	34		
circ-BOK-6	439–609	56	210/255/317	High/High/Very high
	38–442	134		
	114–218	34		
circ-BOK-7	19–123	34	115/160/222	High/High/High
circ-BOK-9	457–630	57	–	–
	92–286	64		
	141–491	116		
circ-BOK-11	39–143	34	135/180/242	High/High/High
circ-BOK-12	525–617	30	277/354/499/544	Moderate/Low/Moderate/Moderate
circ-BOK-13	89–193	34	185/230/292	High/High/High
circ-BOK-14	117–221	34	213/258	High/High
circ-BOK-15	55–159	34	16/203	Moderate/High
	401–844	147		
	525–809	94		
	273–428	51		
circ-BOK-17	301–405	34	147/397/442	Low/High/High
circ-BOK-19	84–290	68	58/103/165	High/High/Very high
circ-BOK-22	127–231	34	223/268	High/Very high

^1^ The prediction of m^6^A sites on the novel *BOK* circRNA sequences was performed with the SRAMP tool. Abbreviations: aa, amino acid; m^6^A, N6-methyladenosine; nt, nucleotide.

**Table 4 ncrna-09-00057-t004:** First-round and nested PCR primer pairs, for the production of nested PCR amplicons from the cDNA of each cell line.

	*BOK* Exon ^1^	Primer Direction	Primer Sequence (5′→3′)
First-round PCR	2nd coding exon	Sense	CTGTGGTGACCGATGCGTTC
		Antisense	TGCAGGGAGATGTGCAGCT
	3rd coding exon	Sense	TGCCTGGGGGAGTTCGT
		Antisense	CCTGCCTCACACAGTCCA
	4th coding exon	Sense	CTTCTTCGTGCTGCTGCCA
		Antisense	CCGAAGCTGCAGAGTGCA
Nested PCR	2nd coding exon	Sense	TGGCTGGCCACATCTTCTC
		Antisense	GCCACGTTGCGGTAGAC
	3rd coding exon	Sense	GCAAGACCCTGGCAACCT
		Antisense	GCATACAGGGACACCACCTT
	4th coding exon	Sense	GAGATGAGCTGCCCACCTG
		Antisense	GCTGACCACACACTTGAGGAC

^1^ The numbering of coding exons was performed based on the annotation of the main *BOK* mRNA (RefSeq ID: NM_032515.5).

## Data Availability

The raw nanopore sequencing reads have been deposited to the Sequence Read Archive (SRA) of NCBI, with BioProject accession numbers PRJNA982806 and PRJNA982812.

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
