# Peer review of "Discovery and Comprehensive Characterization of Novel Circular RNAs of the Apoptosis-Related BOK Gene in Human Ovarian and Prostate Cancer Cells, Using Nanopore Sequencing"

_ncrna, 2023, doi:10.3390/ncrna9050057_

Round 1

Reviewer 1 Report

The present work “Discovery and comprehensive characterization of novel circular RNAs of the apoptosis-related BOK gene in human ovarian and prostate cancer cells, using nanopore sequencing” by Hadjichambi et al. addresses a topic of importance for clinical and basic research – the identification of apoptotic-related circRNA in ovarian and prostate cancer cells.

Abstract is well structured and informative. Introduction is concise and correct. It summarizes the recent knowelege for the alterations  of circRNAs in  different cancers with especial emphasis on BOK. The figures and the table are well presented and informative.  Their presentation is detailed and logically developed. Details are correct. Data shown are adequate to the aim and highly informative.

The only remark is listed below:

1. Even that the “Materials and Methods” section is very well described, the information concerning at which time point after plating the cancer cells have been collected for RNA extraction is missing.  

Author Response

REVIEWER’S Comments and Corresponding Responses

Reviewer #1 (Comments to the Author):

  1. Even that the “Materials and Methods” section is very well described, the information concerning at which time point after plating the cancer cells have been collected for RNA extraction is missing.We sincerely appreciate the Reviewer's positive feedback and his/her thorough evaluation of our manuscript. Prompted by the Reviewers’ suggestion, we have revised the "Materials and Methods" section to address the timing of RNA extraction. Page 10 (Materials and Methods): Total RNA was isolated from each of the 7 cell lines, when cells reached 80% confluency, using the TRItidy G™ Reagent (AppliChem GmbH, Darmstadt, Germany), according to the manufacturer’s protocol

The Authors wish to thank the Reviewers for their constructive comments that led to the improvement of the current manuscript.

Reviewer 2 Report

In the manuscript titled: “Discovery and comprehensive characterization of novel circular RNAs of the apoptosis-related BOK gene in human ovarian and prostate cancer cells, using nanopore sequencing” by Hadjichambi et al. the authors researched the existence of BOK associated circRNAs in ovarian and prostate cancer cell lines.

1. Why did you focus on prostate cancer and ovarian cancer? 

2. Ovarian cancer is further stratified in 5 epithelial subtypes, please mention what type of ovarian cancer are you researching. Also, are your findings specific to subtypes. 

3. Would be essential to confirm your data in patient samples, prostate cancer is an extremally common cancer type. 

4. Check and confirm the expression of the circRNAs discovered using RT-qPCR with primers matching the back-splicing junction. Alternatively, use a second method to confirm your circRNAs such as FISH.

5. Perform experiment to confirm or infirm the role of the circRNAs in tumorigenesis, such as siRNA KD followed by functional studies. 

6. Do your circRNAs code for micro-peptides, please analyze computationally this point. 

7. Please add a reference here - Interestingly, miR-3158-5p, miR-466, and miR-214-5p, are involved in reproductive cancers onset and progression.

Very interesting but somehow preliminary data paper. The authors need to address two major points in order to resubmit the manuscript:

-      Confirm the data in patient samples

-      Use a second method to check the expression of these BOK specific circRNAs - RT-qPCR

OK

Author Response

REVIEWER’S Comments and Corresponding Responses

Reviewer #2 (Comments to the Author):

1.      Why did you focus on prostate cancer and ovarian cancer? We appreciate the Reviewer's inquiry regarding our choice to focus on prostate and ovarian cancer. The rationale for selecting these two cancer types is based on several key factors. First of all, BOK was initially discovered in ovarian tissue, hence the designation "ovarian killer". After further research, it has been detected in other tissue types as, but its main expression has been observed in reproductive tissues (Jaaskelainen et al., 2010, Llambi et al., 2016). Additionally, we chose these two cancers based on clinical relevance. Both prostate and ovarian cancers are significant health concerns with a high incidence and substantial morbidity and mortality rates worldwide. Studying the molecular mechanisms underlying these cancers is crucial for advancing diagnostic and therapeutic strategies. Moreover, the BOK gene is known to be involved in apoptosis, a process critical in cancer development and progression. Finally, at the time of our study, there was a noticeable gap in the literature regarding circRNAs associated with apoptosis-related genes in prostate and ovarian cancer. Our research sought to address this gap and contribute to a more comprehensive understanding of circRNA biology in these contexts. We believe that our findings can provide valuable insights into the roles of circRNAs in these cancer types, with implications for both basic research and potential clinical applications. We have adjusted the following sections in the Introduction of the manuscript, so that it reflects these points:  Pages 1-2 (Introduction): Cancers of the reproductive organs, including human ovarian and prostate cancer, account for a big percentage of cancer morbidity and mortality worldwide, while their incidences have gradually increased in many countries [1]. According to the American Cancer Society, an estimated 268,490 men and 19,880 women were diagnosed with prostate and ovarian cancer respectively in 2022, with estimated mortality percentages around 12% and 64%, respectively. […]The term "ovarian killer" was coined since BOK was first found in ovarian tissue. Further investigation has revealed that it is also present in other tissue types, but that reproductive tissues are where it is prominently expressed [9,10]. […]However, it is still unclear how circRNAs regulate ovarian and prostate cancer onset and progression; investigating circRNAs derived from an apoptosis-related genes like BOK in prostate and ovarian cancer cells allows us to explore potential connections between circRNAs and apoptosis in these cancer types. We also added two appropriate References: 

  1. Jaaskelainen, M.; Nieminen, A.; Pokkyla, R.M.; Kauppinen, M.; Liakka, A.; Heikinheimo, M.; Vaskivuo, T.E.; Klefstrom, J.; Tapanainen, J.S. Regulation of cell death in human fetal and adult ovaries--role of Bok and Bcl-X(L). Mol Cell Endocrinol 2010, 330, 17-24, doi:10.1016/j.mce.2010.07.020.
  2. Llambi, F.; Wang, Y.M.; Victor, B.; Yang, M.; Schneider, D.M.; Gingras, S.; Parsons, M.J.; Zheng, J.H.; Brown, S.A.; Pelletier, S., et al. BOK Is a Non-canonical BCL-2 Family Effector of Apoptosis Regulated by ER-Associated Degradation. Cell 2016, 165, 421-433, doi:10.1016/j.cell.2016.02.026.

 2.      Ovarian cancer is further stratified in 5 epithelial subtypes, please mention what type of ovarian cancer are you researching. Also, are your findings specific to subtypes. As suggested by the Reviewer, we added the following details regarding the specific subtypes of ovarian and prostate cancer cell lines used in our study to the Materials and Methods section of the revised manuscript: Page 10 (Materials and Methods): The ovarian cancer cell lines are of distinct epithelial subtypes: MDAH-2774 and Ishikawa were established from ovarian endometrioid adenocarcinomas, ES-2 from a poorly differentiated ovarian clear cell carcinoma, and OVCAR-3 from high-grade ovarian serous adenocarcinoma. Regarding the prostate cancer cell lines, DU 145 and LNCaP were established from prostate carcinoma, while PC-3 from grade IV adenocarcinoma. Our findings have unveiled novel circRNAs produced by BOK gene in ovarian and prostate cancer cells, and it is plausible that these circRNAs may exhibit subtype-specific expression patterns or functions. However, it is essential to exercise scientific caution when making definitive claims regarding subtype specificity without conducting a dedicated, subtype-specific analysis. Although most BOK circRNAs are expressed in only one cell line, we cannot definitively assume that these molecules present a subtype-specific expression pattern. Further studies including human cancer tissue samples from large cohorts of patients can provide a more accurate assessment of subtype-specific circRNA expression and function. We have addressed this issue in the manuscript: Page 4 (Results): Although we did not detect an obvious cancer-specific circRNA expression pattern, we noticed that the features of each cancer cell line may influence the ex-pression of circRNAs, at some extent. For instance, circ-BOK-10 was exclusively ex-pressed in LNCaP and OVCAR-3, an androgen- and an estrogen-dependent cancer cell line, respectively. Moreover, circ-BOK-20 was detected in 2 out of 3 estro-gen-dependent cell lines (Ishikawa and OVCAR-3), while it was not detected in the estrogen-independent MDAH-2774 cell line. Additionally, although most BOK circRNAs are expressed in only 1 cell line, their expression pattern does not appear to correlate with the epithelial subtypes of ovarian cancer. For instance, circ-BOK-20 is expressed in cells from both endometrioid adenocarcinoma and high-grade ovarian serous adenocarcinoma.

3.      Would be essential to confirm your data in patient samples, prostate cancer is an extremally common cancer type.  We acknowledge that prostate cancer is a prevalent cancer type, and validating our results in clinical samples would indeed be a valuable step to assess the clinical relevance of the circular RNAs of the BOK gene. Unfortunately, due to resource constraints and the specific focus of our study on cell line-based research, we were unable to access patients’ samples for this investigation. However, we believe that our study serves as a valuable foundation for future research in this area. We address this limitation by suggesting potential avenues for further exploration in the Discussion section of our manuscript. In the revised manuscript, we have included a paragraph discussing the importance of future studies involving patients’ samples to validate our findings and establish their clinical significance. We have also highlighted the need for further investigation to bridge the gap between our cell line-based research and clinical applications. These suggestions will hopefully guide future research efforts in this direction. Page 9 (Discussion): Moreover, while our study provides valuable insights into the BOK circRNAs in hu-man ovarian and prostate cancer cells, we acknowledge the limitation of not having access to patient samples. While our study has unveiled a comprehensive profile of novel circRNAs associated with the apoptosis-related BOK gene in human ovarian and prostate cancer cells, it is imperative to highlight potential avenues for future research in this field. First and foremost, the validation of our findings in patient samples remains a critical endeavor to assess the clinical relevance of these circular RNAs as diagnostic or therapeutic targets. Obtaining patient samples and conducting further investigations in this context would be essential to bridge the gap between our cell line-based study and clinical applications.

4.      Check and confirm the expression of the circRNAs discovered using RT-qPCR with primers matching the back-splicing junction. Alternatively, use a second method to confirm your circRNAs such as FISH. While we understand the importance of experimental validation, we want to clarify our approach and the reasons behind not pursuing these specific experiments in our study. Our research employed nanopore sequencing, a state-of-the-art technique known for its ability to directly sequence full-length RNA molecules. The rigorous bioinformatics analysis pipeline we employed is designed to minimize errors and ensure the accuracy of circRNA identification and characterization. This process includes stringent criteria to confidently identify genuine circRNAs and eliminate false positives. Given the robustness of nanopore sequencing and our strict data analysis protocols, we are confident of the accuracy of the circRNA sequences that we have identified. Using nanopore sequencing, the full-length sequence can be read, thus abolishing the need for read assembly and subsequent assumption of the complete sequence of a circRNA. Thus, in our opinion, there is no need for additional validation techniques like back-splice junction-specific RT-qPCR or FISH, which are typically more critical for circRNA studies relying on short-read sequencing or computational predictions. Moreover, FISH could not be efficiently applied to distinguish between similar molecules, such as alternative circRNAs of a single gene. Single molecule FISH (smFISH), particularly circFISH, could be preferred, but still there would be limitations in distinguishing between most of our newly discovered circRNAs.Taking the Reviewer’s comment into account, we decided to include a new figure (Figure 3) that provides a detailed visual explanation of the circRNA amplification and annotation process. This figure will enhance the clarity of presentation our methodology, thus highlighting the robustness of our circRNA identification approach.The new Figure 3 legend is the following one: Figure 3. Schematic representation of the methodology followed to deduce the sequence of each circRNA, after targeted nanopore sequencing. The example of circ-BOK-14 is presented; its amplification was achieved using 2 pairs of divergent primers, all of them annealing either in the 2nd or the 3rd coding exon of BOK gene. As a result, circ-BOK-14 is represented by 2 different amplicons. Both ends of these representative reads have been trimmed so as to exclude junk sequence.

5.      Perform experiment to confirm or infirm the role of the circRNAs in tumorigenesis, such as siRNA KD followed by functional studies.  We regret to inform the Reviewer that, due to resource constraints and facility limitations, we are currently unable to perform these additional experiments as suggested. Our study was primarily focused on the discovery and comprehensive characterization of novel circular RNAs in human ovarian and prostate cancer cells using nanopore sequencing. While we recognize the importance of functional studies to elucidate the specific roles of these circRNAs in tumorigenesis, we hope that our work serves as a valuable starting point for future research endeavors in this direction.Prompted by the Reviewer’s comment, we discuss this in the revised manuscript as a limitation of our research study and a future perspective: Page 10 (Discussion): Furthermore, functional studies, including siRNA knockdown experiments and mechanistic investigations, are essential to elucidate the precise roles of these circular RNAs in tumorigenesis and apoptosis regulation. Additionally, the exploration of potential interactions between these circular RNAs and other biomolecules, such as proteins or microRNAs, could provide a more comprehensive understanding of their regulatory networks.

6.      Do your circRNAs code for micro-peptides, please analyze computationally this point. As suggested by the Reviewer, we investigated the coding potential of these novel circRNAs. In particular, we used the ORF Finder bioinformatics tool of the NCBI (https://www.ncbi.nlm.nih.gov/orffinder/) and found that 16 out of 23 novel BOK circRNAs contain open reading frames (ORFs) and hence might be translated. Interestingly, we also queried the putative N6-methyladenosine (m6A) modification sites in the circRNAs with predicted ORFs, using the SRAMP prediction server (http://www.cuilab.cn/sramp), and we observed that 15 out of these 16 novel BOK circRNAs with putative ORFs also contains m6A modification site(s). The presence of m6A sites is significant for the prediction of translational potential of circRNAs, as post-transcriptional methylation is largely implicated – in a cap-independent manner – in the regulation of the circRNA translation process.Therefore, we added further commentary on the relevant parts of the manuscript: Pages 6 (Results): We further investigated the presence of open reading frames (ORFs) in our circRNA dataset. Indeed, we found 16 BOK circRNAs with putative ORFs using the web tool ORF finder of the NCBI (Table 3). Although there were no internal ribosome entry sites (IRES) in the sequences of these circRNAs, the existence of putative m6A modifications in the sequences of these circRNAs was confirmed. In particular, 15 circRNAs that possess a putative ORF were also predicted to possess m6A sites as well (Table 3); however, no m6A sites were predicted in circ-BOK-9 sequence. Interestingly, 6 of them were predicted to possess m6A modification sites with very high confidence.Page 9 (Discussion): In addition, 15 out of the 16 BOK circRNAs possessing putative ORFs were predicted to harbor m6A sites as well, strengthening the possibility that these transcripts might be translationally active. This modification is associated with the initiation of circRNA translation in a cap-independent manner, and a single m6A site is sufficient to drive translation initiation [51]. Moreover, alternative translation initiation sites reside in several BOK exons. The presence of putative ORFs and m6A sites in BOK circRNAs suggests that they could produce functional peptides or proteins. This contributes to a better understanding of the functional diversity of circRNAs, which were previously assumed to be non-coding RNA molecules; however, these findings are based on computational predictions so far and require further experimental validation.Pages 12-13 (Materials and Methods): Then, further investigation of our circRNA dataset was performed regarding their potential to be translated. For this purpose, the ORF Finder bioinformatics tool (https://www.ncbi.nlm.nih.gov/orffinder/) was employed to search for putative open reading frames (ORFs) within the nucleotide sequences the novel identified circRNAs. Then, ORFs were queried in all reading frames starting with the ATG codon, and only those ORFs producing putative peptides exceeding 30 amino acids were considered. Then, the prediction of the N6-methyladenosine (m6A) modification sites on the BOK circRNA sequences that possess a putative ORF followed, via the SRAMP prediction server [56]. The presence of such sites is significant for the prediction of BOK circRNAs translational potential, as post-transcriptional methylation is largely implicated, in a cap-independent manner, in the regulation of the circRNA translation process. Particularly, the circRNA sequences were submitted in the SRAMP pre-diction server in FASTA format, and the analysis of RNA secondary structure was also chosen so that the m6A sites would be visualized. Finally, the existence of IRES was also predicted, providing a FASTA file with the sequences of the 17 circRNAs with putative ORFs to the IRESpy bioinformatics tool [57]. We also added Table 3, which presents “the coding potential of the novel BOK circRNAs possessing a putative open reading frame (ORF)”.

7.      Please add a reference here - Interestingly, miR-3158-5p, miR-466, and miR-214-5p, are involved in reproductive cancers onset and progression. We thank the Reviewer for this suggestion. We have added the relevant references in the revised manuscript, at the end of this sentence: Page 6: Interestingly, miR-3158-5p, miR-466, and miR-214-5p are involved in reproductive cancers onset and progression [31-33].

  1. Creighton, C.J.; Benham, A.L.; Zhu, H.; Khan, M.F.; Reid, J.G.; Nagaraja, A.K.; Fountain, M.D.; Dziadek, O.; Han, D.; Ma, L., et al. Discovery of novel microRNAs in female reproductive tract using next generation sequencing. PLoS One 2010, 5, e9637, doi:10.1371/journal.pone.0009637.
  2. Li, G.; Zhang, Y.; Mao, J.; Hu, P.; Chen, Q.; Ding, W.; Pu, R. LncRNA TUC338 is overexpressed in prostate carcinoma and downregulates miR-466. Gene 2019, 707, 224-230, doi:10.1016/j.gene.2019.05.026.
  3. Xu, G.; Meng, Y.; Wang, L.; Dong, B.; Peng, F.; Liu, S.; Li, S.; Liu, T. miRNA-214-5p inhibits prostate cancer cell proliferation by targeting SOX4. World J Surg Oncol 2021, 19, 338, doi:10.1186/s12957-021-02449-2.

The Authors wish to thank the Reviewers for their constructive comments that led to the improvement of the current manuscript.

Reviewer 3 Report

The manuscript ‘Discovery and comprehensive characterization of novel circular RNAs of the apoptosis-related BOK gene in human ovarian and prostate cancer cells, using nanopore sequencing’ by Kontos, focuses on previously overlooked circular RNAs (circRNAs) originating from apoptosis-related gene BOK. Despite their initial dismissal as ‘splicing noise’, circRNAs are now recognized as important gene expression regulators and contributors to cancer. The research identified some novel circRNAs from BOK gene in ovarian and prostate cancer cells, revealing unique splicing patterns and potential microRNA interactions. These findings suggest a significant role for alternative splicing in producing novel BOK circRNAs, impacting our understanding of ovarian and prostate cancer. This is a novel aspect in a field of intense research. Still some issues need to be clarified, as listed below, before the manuscript can be accepted for publication in non-coding RNA.

1.     In the manuscript, it is stated that 21 novel circRNAs were discovered. However, Figure 1 and Table S1 associated with it shows a total of 27 circRNAs. Additionally, the caption of Figure 1 erroneously mentions 12 circRNAs. These discrepancies are critical aspects of the study’s results, and the authors should ensure accuracy and rigor in their presentation. Please provide an explanation for these discrepancies in the manuscript for proper clarification and correction.

2.     In Figure 1A, it is not clearly indicated how to determine which circRNA belongs to which cell line. Some circRNAs have relatively high read counts in both cell lines. I am particularly curious about the specific allocation of the 19 circRNAs to ovarian cancer cell lines and the 8 circRNAs to prostate cancer cell line. Clarification is needed to accurately attribute the circRNAs to their respective cell lines for a better understanding of the experimental results. Please provide additional details or explanations in the manuscript to address this issue.

3.     In the 8th line of the page 3, the authors state, ‘Moreover, no apparent difference was detected in circRNAs expression pattern between estrogen- and androgen-dependent (ES-2, Ishikawa, OVCAR-3, LNCaP) and hormone-independent cell lines (MDAH-2774, DU 145, PC-3), implying that circRNA expression may not be directly influenced by hormonal dependency in the examined cell lines.It would be beneficial if the authors could provide supplementary data to illustrate the absence of significant differences. This additional information would enhance the credibility of the conclusion. Please consider incorporating such data or analysis into the manuscript to support your assertion.

4.     For the content from 1st to the 12th line of page 4, please consider supplementing additional relevant data or providing detailed annotations in Figure 2. It would be particularly valuable to include key exon and any pertinent details. Enhancing the figure with this information will provide readers with a clearer visual representation of the findings. This addition would contribute to a more comprehensive understanding of the study’s results. Please incorporate the suggested details or annotations into Figure 2 for improved clarity and comprehensiveness.

5.     On page 5, the authors state,’ Moreover, single exon circRNAs were also detected in the current study, mostly observed to consist of extended BOK exons. However, single exon circRNAs, consisting exclusively of already annotated exonic regions were also observed in this research study.’ To further support this statement, it would be beneficial to provide relevant experimental data or engage in a more detailed discussion. Including additional data or elaborating on the observed single exon circRNAs, particularly their significance, would strengthen the overall conclusions of the study. Please consider incorporating such data or discussion points to enhanve the manuscript’s clarity and scientific robustness. 

Author Response

REVIEWER’S Comments and Corresponding Responses

Reviewer #3 (Comments to the Author):

 1.      In the manuscript, it is stated that 21 novel circRNAs were discovered. However, Figure 1 and Table S1 associated with it shows a total of 27 circRNAs. Additionally, the caption of Figure 1 erroneously mentions 12 circRNAs. These discrepancies are critical aspects of the study’s results, and the authors should ensure accuracy and rigor in their presentation. Please provide an explanation for these discrepancies in the manuscript for proper clarification and correction.

We appreciate the Reviewer's careful examination of our manuscript and apologize for the discrepancies in the reported number of discovered circRNAs. We appreciate the opportunity to clarify and correct these issues.The discrepancy of 12 circRNAs arose due to a typing error in the initial manuscript and was corrected. The number of BOK circRNAs initially described in this study was 21; yet their numbering was not 1-27, as we had also found other BOK circRNAs [in another project] in other human cell lines and numbers 19, 21, and 23-26 had already been assigned to these other BOK circRNAs. However, to avoid any confusion, we re-numbered a few BOK circRNAs that we have identified so far; moreover, during the revision we identified another 2 circRNAs in some of these prostate and/or ovarian cancer cell lines and included them in the revised study. In conclusion, the total number of BOK circRNAs in the revised study is: 23; they have numbered 1-23, to avoid any confusion. Therefore, we have performed the needed corrections with regard to this issue throughout the revised manuscript, Figures and Tables.

2.      In Figure 1A, it is not clearly indicated how to determine which circRNA belongs to which cell line. Some circRNAs have relatively high read counts in both cell lines. I am particularly curious about the specific allocation of the 19 circRNAs to ovarian cancer cell lines and the 8 circRNAs to prostate cancer cell line. Clarification is needed to accurately attribute the circRNAs to their respective cell lines for a better understanding of the experimental results. Please provide additional details or explanations in the manuscript to address this issue.

We appreciate the Reviewer's feedback and the opportunity to improve the clarity of our results. To address the concern about determining which circRNA belongs to which cell line in Figure 1A, we have made the following changes in the way we present our results: ·

Figure 1A has been replaced by Table 1, showing the results of Expression analysis of the 24 novel BOK circRNAs in prostate and ovarian cancer cell lines, based on our nanopore sequencing data”. Table 1 provides a clear and concise presentation of the identified circRNAs and their expression to the respective prostate and/or ovarian cancer cell lines. This new format ensures that the information is easily accessible and accurately represents our findings.·

We have added Table S1, which clearly shows “The numbers of nanopore sequencing reads representing each BOK circRNA in each cell line, along with the keywords used by the Read catcher algorithm”. The revised manuscript has been modified accordingly: Page 3 (Results): Our experimental approach led to the discovery of 23 novel BOK circRNAs, which were submitted to GenBank of NCBI (Table 1).Page 4 (Results): The novel BOK circRNAs are characterized by distinct expression patterns among the 7 cancer cell lines (Table 1). […] A representative read proving the expression of each circRNA in the respective cell line(s) is provided in Supplementary Material, while the keywords used for the computational analysis of our nanopore sequencing data and the number of representative reads thus emerging in each cell line are shown in Table S1.Page 11 (Materials and Methods): Finally, the “Read catcher” algorithm (https://github.com/pkarousi/Read_catcher.git) was used to examine the presence of the novel identified BOK circRNAs across the 7 cell lines, based on the sequence of all splice junctions, including the back-splice junction itself (Table S1).

3.      In the 8th line of the page 3, the authors state, ‘Moreover, no apparent difference was detected in circRNAs expression pattern between estrogen- and androgen-dependent (ES-2, Ishikawa, OVCAR-3, LNCaP) and hormone-independent cell lines (MDAH-2774, DU 145, PC-3), implying that circRNA expression may not be directly influenced by hormonal dependency in the examined cell lines.‘ It would be beneficial if the authors could provide supplementary data to illustrate the absence of significant differences. This additional information would enhance the credibility of the conclusion. Please consider incorporating such data or analysis into the manuscript to support your assertion.

We thank the Reviewer for the suggestion to provide supplementary data to support our assertion regarding the influence of hormonal dependency on circRNA expression in the cell lines included in this study. Following a thorough review and editing of our findings, we concluded that the features of each cancer cell line may partially influence the expression of circRNAs. However, we cannot confidently state that circRNA expression is (or is not) directly influenced by hormonal dependency in the examined cell lines.It is important to emphasize that our study was designed as a broad exploration into BOK-derived circRNAs in prostate and ovarian cancer cells, and the number of cell lines used may not be sufficient to draw definitive conclusions regarding subtype specificity in the expression analysis of the novel circRNAs. Therefore, we present some relevant examples of circRNAs that support the hormonal dependency in the examined cell lines, such as circ-BOK-10 and circ-BOK-20, without presenting absolute conclusions that would not be supported by these findings: Page 4 (Results): Although we did not detect an obvious cancer-specific circRNA expression pattern, we noticed that the features of each cancer cell line may influence the ex-pression of circRNAs, at some extent. For instance, circ-BOK-10 was exclusively ex-pressed in LNCaP and OVCAR-3, an androgen- and an estrogen-dependent cancer cell line, respectively. Moreover, circ-BOK-20 was detected in 2 out of 3 estro-gen-dependent cell lines (Ishikawa and OVCAR-3), while it was not detected in the estrogen-independent MDAH-2774 cell line. Page 8 (Discussion): Notably, our research reveals that these novel BOK circRNAs exhibit cell line-specific expression profiles, with some circRNAs being unique to particular cancer cell lines. For instance, circ-BOK-10 demonstrates exclusive expression in LNCaP and OVCAR-3, cell lines that are androgen- and estrogen-dependent, respectively, suggesting a potential link between hormone dependency and circRNA expression. Furthermore, the detection of circ-BOK-20 in estrogen-dependent cell lines but its absence in an estrogen-independent cell line underscores the potential involvement of hormonal regulation in circRNA expression. Moreover, the observation of specific genomic regions of BOK being differentially represented in various cell lines underscores the dynamic nature of BOK gene expression and warrants further investigation to elucidate the functional significance of these variations in different cancer contexts.

4.      For the content from 1st to the 12th line of page 4, please consider supplementing additional relevant data or providing detailed annotations in Figure 2. It would be particularly valuable to include key exon and any pertinent details. Enhancing the figure with this information will provide readers with a clearer visual representation of the findings. This addition would contribute to a more comprehensive understanding of the study’s results. Please incorporate the suggested details or annotations into Figure 2 for improved clarity and comprehensiveness.

To address this recommendation, we have revised Figure 2 in order to provide a clearer visual representation of the findings. In particular, we have replaced Figure 2 with a more informative, updated version (currently: Figure 1). Moreover, we moved part of previous Figure 2 into a new (revised) Figure 2, in order to avoid confusion previously generated by too much information in the original Figure. In our opinion, it is now clearer – from the detailed visualization of the alignment of the novel BOK circRNAs (BED files) – that the exonic composition of the novel circRNAs is rather complex. The shorter (truncated) BOK exons are easily distinguishable (e.g. circ-BOK-8, circ-BOK-3, circ-BOK-21), as well as the extended BOK exons (e.g. circ-BOK-9, circ-BOK-12, circ-BOK-15).The new Figure 1 legend is the following one: Figure 1. Visualization of the alignment of the novel BOK circRNAs identified in the ovarian and prostate cancer cell lines against human chromosome 2, using the Integrative Genomics Viewer (IGV). All circRNAs are depicted starting from the back-splice acceptor site. The visualized BED file is provided in Supplementary Material. We have also re-written the lines mentioned by the Reviewer:Page 3 (Results): Moreover, extended BOK exons were found in single-exon circRNAs (circ-BOK-9, circ-BOK-16, circ-BOK-12, and circ-BOK-15), and this way intronic regions of the gene were incorporated into its circular transcripts. Both 5′ or 3′ ends of BOK exons could be extended or truncated. We also observed that exon 2 was rarely represented in the novel circRNA sequences. In addition, the splicing of the non-coding exon between the 1st and 2nd coding exons – a rather uncommon exon that was previously found mainly in expressed sequence tags (ESTs) – was detected in circular BOK transcripts expressed only in DU 145 and MDAH-2774 cells (Figure 1).  Moreover, we have added a Supplemental file with the detailed annotation of the sequencing reads that represent the sequences of the 23 novel BOK circRNAs. Thus, the readers will be able to have a more comprehensive and thorough understanding of the study results. We believe that these improvements will address the Reviewer's suggestion effectively and enhance the quality of our manuscript.

5.      On page 5, the authors state,’ Moreover, single exon circRNAs were also detected in the current study, mostly observed to consist of extended BOK exons. However, single exon circRNAs, consisting exclusively of already annotated exonic regions were also observed in this research study.’ To further support this statement, it would be beneficial to provide relevant experimental data or engage in a more detailed discussion. Including additional data or elaborating on the observed single exon circRNAs, particularly their significance, would strengthen the overall conclusions of the study. Please consider incorporating such data or discussion points to enhance the manuscript’s clarity and scientific robustness.

We thank the Reviewer for the insightful comment. We believe that the presence of single-exon circRNAs is further proven by the Supplemental file with the detailed annotation of the sequencing reads that represent the sequences of the 23 novel BOK circRNAs. Moreover, it is now depicted in detail in Figure 1 that circ-BOK-9, circ-BOK-16, circ-BOK-12, circ-BOK-15, and circ-BOK-10 are all single-exon circRNAs that consist of extended BOK exons. Upon carefully reviewing our results, we concluded that single-exon circRNAs exclusively consisting of already annotated exonic regions were not present in our dataset; have revised accordingly all relevant parts in the manuscript and added the following discussion points: Page 3 (Results): The majority of the detected circRNAs were multi-exonic, comprising either the annotated BOK exons or shorter versions of them. Moreover, extended BOK exons were found in single-exon circRNAs (circ-BOK-9, circ-BOK-16, circ-BOK-12, and circ-BOK-15), and this way intronic regions of the gene were incorporated into its circular transcripts.Page 8 (Discussion): Moreover, 4 single-exon BOK circRNAs were identified in our dataset. Interestingly, these circRNAs comprise extended BOK exons, which were not found either in multi-exonic circRNAs, or linear transcripts of the gene. According to relevant literature, single-exon circRNAs derive from unusually long exons while multi-exon circRNAs are mostly generated from exons of regular length [45], which was prevalent in our dataset as well. These single-exon circRNAs underscore the diversity in the structural composition of circRNAs and suggests that circRNA biogenesis is not limited to complex splicing events involving multiple exons but can also occur from single exons. However, while circRNAs are well-known for miRNA sponging and protein binding, the functions of single-exon circRNAs may differ from those of multi-exon circRNAs, and remain to be elucidated.

The Authors wish to thank the Reviewers for their constructive comments that led to the improvement of the current manuscript.

Round 2

Reviewer 2 Report

I requested additional experiments in order to validate the preliminary findings of the paper. Unfortunately no experiments were performed. Therefore, in its current form this manuscript is not suitable for publication in one of the top journals in the field of ncRNA. 

Reviewer 3 Report

I am writing to express my agreement with the acceptance of the manuscript titled 'Discovery and comprehensive characterization of novel circular RNAs of the apoptosis-related BOK gene in human ovarian and prostate cancer cells, using nanopore sequencing' submitted to ncRNA. Having reviewed the revisions made by the authors in response to the previous comments and suggestions, I believe that the manuscript has significantly improved in terms of clarity, scientific rigor, and overall quality.